# A Land Administration Data Exchange and Interoperability Framework for Kenya and Its Significance to the Sustainable Development Goals

Clifford Okembo [1,2,*], Javier Morales [1], Christiaan Lemmen [1], Jaap Zevenbergen [1] and David Kuria [3]

1 Faculty of Geo-Information Science and Earth Observation (ITC), University of Twente, P.O. Box 217, 7500 AE Enschede, The Netherlands; j.morales@utwente.nl (J.M.); c.h.j.lemmen@utwente.nl (C.L.); j.a.zevenbergen@utwente.nl (J.Z.)
2 Department of Geosciences and Environment, School of Physics and Environment, Technical University of Kenya, P.O. Box 52428, Nairobi 00200, Kenya
3 Land Administration and Management Directorate, National Land Commission, P.O. Box 44417, Nairobi 00100, Kenya; dn.kuria@gmail.com
* Correspondence: c.o.okembo@utwente.nl; Tel.: +254-723-525-944

**Abstract:** Sharing land data from one department to the other is a continuous process. A solid structure and a set of guidelines on how to share them is to be put in place as a foundation for the development of a land administration data exchange and interoperability framework in support of data acquisition, land transactions and distribution of land data. In this research, the application of the ISO Framework for Enterprise Interoperability (FEI) as a standard is the starting point. Utilising the Land Administration Domain Model (LADM) profile for Kenya as a base, an interoperability framework in support of land administration in Kenya is developed that addresses concerns, removes barriers and selects the approach for implementation. Due to the critical nature of land, it fits into the United Nations 2030 sustainability agenda. During the development of the Kenyan profile, four country-specific issues in the context of people-to-land relationships have been identified and modeled. The mapping of those issues relevant to the sustainable development goals supports the achievement of those goals so that all related targets and indicators can be attained. Using GIS tools, the implementing and testing of the new LADM profile for Kenya is not a difficult task. By using existing land data combined with newly collected data in the LADM-compliant database, a complete and accurate workflow is assured. Integration with external databases is useful for improving efficiency and eliminating duplication. Data collection with all stakeholders and validation through public inspection are recommended.

**Keywords:** country profile; data exchange; FEI; interoperability; Kenya; LADM; SDGs

## 1. Introduction

Kenya is at the initial stages of developing its land information management system called *Ardhisasa* (Swahili word for *land now*). The system was launched in 2021 in support of the Nairobi land registry [1]. It is expected to be developed fully for nationwide coverage and usage, providing support to several functions, including land taxation, spatial planning and tenure security, with workflows for initial data acquisition, establishment of rights, land surveying, parcel subdivision and information provision. It is prepared for use by several stakeholders such as registrars, surveyors, conveyors, valuation and spatial planning professionals, the government at both national and decentral level and the public in general. However, the system has not yet been in full use due to the need for some improvements in functionalities experienced while using it; see [2,3]. Conversion of land data from an analogue to digital data environment is ongoing.

Different organisations and departments dealing with land administration and land management do not have a framework for sharing their produced data with other stakeholders directly or indirectly. As was already observed by Wayumba (2013), this hinders the quick cross-referencing of records and constrains the orderly and timely updates of databases in use. In practice, this results in separated data siloed in different locations with access constrains and data duplications [4]. As much as there have been efforts in implementing *Ardhisasa*, it has not yet climaxed in the development of a comprehensive cadastral model as the parcel information contained in the existing model is only geometry information from the traditional cadastre without attributes [5]. This means that the system only has some basic cadastral information that is not linked with the registry data.

The above indicates the need for a system that allows all users to create, add, access and manipulate data in a manner that serves their purposes efficiently, hence the need for data exchange within the system. For data exchange to be effective, there must be a standard and shared data model that is agreed upon and used by all the stakeholders involved in the collection, maintenance and distribution of the land data [6]. Putting usable standards in place is believed to reduce data exchange costs [7,8]. In this case of land data, a Land Administration Domain Model (LADM) profile for Kenya is developed to facilitate this [1]. The model should function as fully coordinated and automated, without separation of data from the land registry and cadastre [9], while keeping their organisational mandates intact. The two should not be the limit but should allow for the access of other actors of land administration too: local government, environmental, infrastructure, spatial planning and taxation agencies and utilities, as well as the construction sector and the actors of the real estate market [7]. The model could also accommodate the linkages of the mining cadastre with the land registry and cadastre [9].

The integration of the land system with other external data sources, such as the judiciary, tax authorities, such as the Kenya Revenue Authority (KRA), registrar of companies and land-buying companies (brokers and investors) [4] is important [10]. This could be similar to the integration of the key registers in the Netherlands [11], as shown in Figure 1. Both geographical and non-geographical (administrative) data are defined to facilitate information exchange, which is important between the government, citizens and businesses [12]. The linkages are determined by the information requirements of one entity to the other; for example, for vehicle registration, a resident's address is required. However, it could have the functionality to distinguish the different institutions' tasks and mandates and to align them with each other, all on a conceptual level [7].

Standards and standardisation are among the foundational building blocks for interoperability for digitalisation, and standardisation of data is recommended so as to allow for data interoperability [6,13–15]. This could be in the form of a data model, data formats, system design and implementation. For the land sector, the challenge of interoperability can only be addressed if the land information systems are based on a common data model [16]. The LADM provides a good and standards-based model that can be used to improve land administration through the implementation of an integrated land information management system (LIMS) [17]. It provides a good starting point to develop land information systems that are interoperable across government departments responsible for land administration and also across different administrative units [16].

This paper is part of a larger research looking at the land administration system in Kenya. The research starts with considering the requirements for the development of a Land Administration Domain Model (LADM) profile for Kenya [4], then uses those requirements to develop an LADM profile for Kenya [1]. The next phase covered by this paper is looking into the implementation of the model and specifically focuses on the data exchange and interoperability framework necessary for the implementation and operation of the profile for Kenya, with the aim of contributing to the achievement of the sustainable development goals (SDGs). This goal is to be achieved by answering the following questions: (1) What are the data exchange and interoperability user requirements in Kenyan LAS? (2) Who are the involved LAS data stakeholders? (3) How is interoperability achieved for Kenya?

(4) How does the Kenya LAS support contributing to the achievement of the SDGs? (5) Does the Kenyan LADM country profile function in real-world implementation?

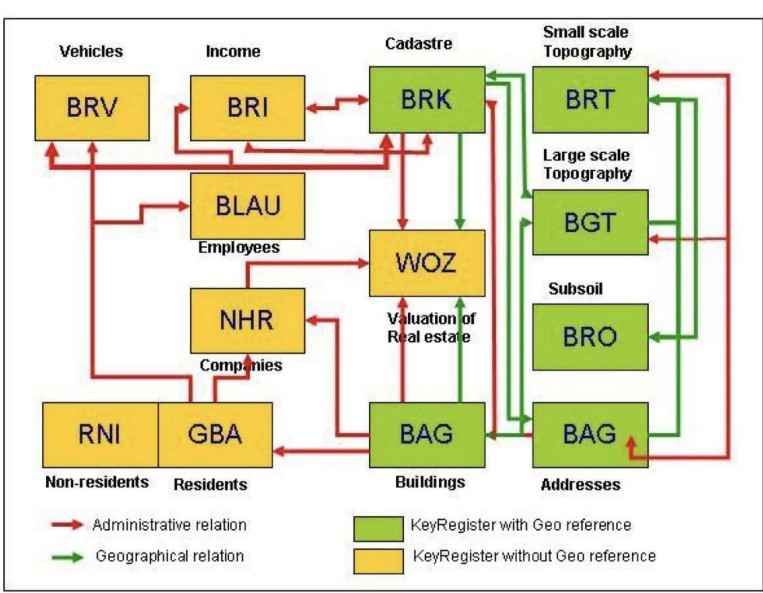

**Figure 1.** Dutch system of key registers (Source: [12]).

This paper is organised into eleven sections, starting with the introduction to the paper in Section 1. Section 2 discusses the methodology used in this paper, and sources of data for exchange and interoperability are summarised in Section 3, while Section 4 presents the interoperability requirements for Kenya. The interoperability framework in land administration for Kenya is discussed in Section 5 and the mapping of Kenya's unique LADM requirements with the SDGs is presented in Section 6. A technical test for the implementation of the LADM Kenya profile [1] was conducted, and the results are presented in Section 8, discussions in Section 9, and conclusion and recommendation in Section 10.

## 2. Methodology

This study employs a field-testing methodology where the model as developed earlier [1,4] shall be put to use by stakeholders and the observations are recorded. In the field-testing methodology, practitioners are provided with the tools based on the model developed to be used for their real-world work through the usual processes to achieve the results [18]. Figure 2 visualises this methodology.

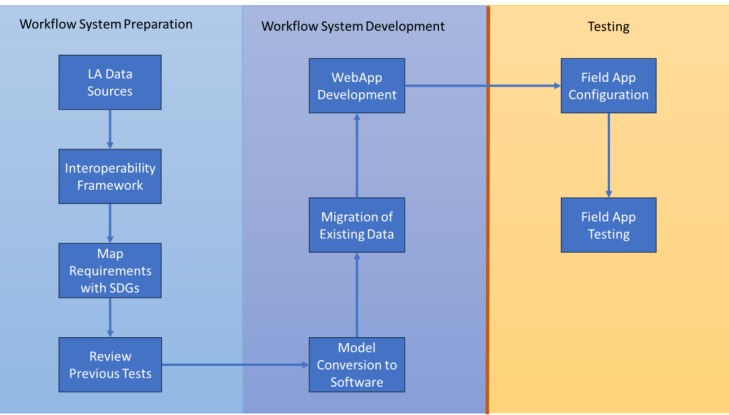

**Figure 2.** Methodology for the study.

The field-testing methodology adopted for this project is explained as follows:

1.  Land administration data sources: the first stage concerns the mapping of all the LA data sources with the authorities responsible for their creation and management. This includes not only the land departments or cadastre and registry, but also the other external data sources necessary for LA functions: land tenure, land use, land value and land development [19]. This stage is discussed in Section 3.

2.  Interoperability framework for Kenya: following the "interoperability guidelines" for LA [14] that are based on the ISO standard [20] on the framework for enterprise interoperability, an interoperability framework for Kenya is developed. This stage is discussed in Sections 4 and 5.

3.  Map the requirements with the SDGs: in this stage, Kenya's unique requirements for the country's LADM profile [4] will be reviewed and mapped to the SDGs and indicators following the sub-workflow in Figure 3.



**Figure 3.** Workflow for mapping Kenya LADM unique requirements with SDGs.

At this stage, the unique Kenyan LADM issues and requirements are identified and reviewed. This will help to identify the SDGs addressing land and land-related issues. The identified SDGs are then mapped to the Kenyan LADM issues identified. The significance of these SDGs to Kenya and the UN Agenda 2030 is discussed. This stage is discussed in Section 6.

4.  Review the previous tests: several tests have been conducted on implementing the LADM. Makueni County in Kenya performed a technical test on the Social Tenure Domain Model (STDM) based on a fit-for-purpose (FFP) approach in 2017 [21] and a recent one in Colombia on the LADM [22]. This gives this test a perspective of the success and challenges faced that are to be considered for improvement. This stage is discussed in Section 7.

5.  Deploy the Kenyan profile into a geographic information system (GIS) software system: the LADM profile for Kenya developed [1] is based on the Unified Modeling Language (UML) that is translated to any database management system (DBMS) preferred. The model is converted into a GIS system schema in GIS software. This step makes the model ready for implementation and use in the other steps as discussed in Section 8.2.

6.  Migrate existing data: there exist parcel data that are available on a digital format. In this stage, the existing data are migrated and entered into the database based on the LADM profile for Kenya as discussed in Section 8.3.

7.  Develop a web application based on the Kenyan profile: web services are necessary for the sharing of information to a wider audience through the Internet [23]. Those services can enable the collaboration of different people involved in land administration processes in different locations and roles. At this stage, a web application is developed to facilitate access of information to all stakeholders in the field and those in offices as discussed in Section 8.4.

8.  Configure the field app tool based on the Kenyan profile: a field data collection tool is developed within an application with a simple and easy-to-use interface. The Colombian case recommends the Field Maps (formerly Collector) app [22], and this is used for this second test in Kenya too. This stage is discussed in Section 8.5.

9.  Field app testing: field data collection is partly based on the experiences from 2017 in Makueni County [21] and partly from Colombia in 2021 [22]. The collection of the identities of the landowners is performed through photographs and digital signatures and fingerprints. This is combined with an actual boundary delineation and entry of the related administration data. This stage includes activities of grassroots surveyors, property owners, professional surveyors, officers from land registry, land administration offices and local security and administration officers like area chiefs. This stage is discussed in Section 8.6.

## 3. Sources of Data for Exchange and Interoperability

Both the cadastre department under the Survey of Kenya and the land registry generate most of the core data for Kenya's LA through their offices and workflows. They register these data both in the field during survey and adjudication work and in the offices during registration processes. Table 1 presents the data collected, entered and managed by these departments.

**Table 1.** Data authorities.

| Land Registry | Directorate of Surveys (Survey of Kenya) |
|---|---|
| - Owner's details<br>- Owner's address<br>- Parcel ID | - Parcel ID<br>- Parcel Size<br>- Survey Plan<br>- Coordinates |

For efficiency in data collection and for the avoidance of errors in data collection while eliminating duplication of records, for example, when adding data about a party, a common key like a national ID number [10] can be entered, and then the rest of the information, such as name, gender, date of birth and address, are auto-populated from the Registrar of Persons database. The Kenyan LADM profile recommends and models for the integration of external data sources. These are other actors when linking to the external classes in the LADM profile for Kenya [1] as presented in Table 2.

**Table 2.** External data authorities.

| Organisation Name | Data Originated |
|---|---|
| Construction Authority of Kenya (CAK) | - Building registration number |
| Kenya Revenue Authority (KRA) | - Tax registration number |
| Registrar of Persons | - Birth registration number<br>- National identification number<br>- Death registration number |
| Registrar of Companies | - Company; limited, NGO, registration |
| Registrar of Societies and associations | - Societies registrations<br>- Places of worship<br>- Associations: professional and social |

**Table 2.** *Cont.*

| Organisation Name | Data Originated |
| --- | --- |
| County Governments | - Valuation rolls<br>- Development controls<br>- Spatial/physical planning<br>- Land rate payments |
| Communication Authority (CA) | - Physical address (yet to function) |
| Postal Corporation of Kenya (Posta) | - Postal numbers<br>- Post codes |

## 4. Requirement for Interoperability in Kenya

Land data exchange and interoperability are a great concern and need in Kenya [13,15]. The government of Kenya is striving to develop *Ardhisasa.* However, it is important to note that the Kenyan cadastre in its current status may not be readily suitable for a computerised environment. This is because the different cadastral maps cannot be readily integrated to create a homogeneous and seamless digital cadastre. This was observed already in [24], where the maps are still kept in manual format.

The government of Kenya is currently focused on the promotion of ease of doing business (EODB) and data interoperability, as well as the introduction of e-governance [13]. This is to enable the achievement of the sustainable development goals (SDGs) and Kenya's Vision 2030 goal, which aims to make Kenya a globally competitive and prosperous country with a high quality of life by 2030 [25]. Kenya formulated and amended several laws, with land-related ones included (though still in draft state): Land Registration (Electronic Transactions) Regulations, 2020; Survey (Electronic Cadastre Transactions) Regulations, 2020; Stamp Duty (Valuation) Regulations, 2020; Stamp Duty (Amendment) Regulations, 2020; Land (Amendment) Regulations, 2020; Land (Extension and Renewal of Leases) (Amendment) Rules, 2020; The Land (Allocation of Public Land) (Amendments, Regulations), 2020; and Physical and Land Use Planning (Electronic Development Control and Enforcement System) Regulations, 2020, to facilitate the digital services provision.

According to the Ministry of Land, Public Works, Housing and Urban Development's (MLPWHUD) (2020) report on electronic land transactions, it is important to align e-government initiatives within an elaborate enterprise system architecture/infrastructure with a holistic view of the interoperability of the developed systems (both public and private). This ensures that digitisation and e-conveyancing can align to such strategies since they are bound to inform or affect other government functions [26]. It is noted that data integration and data interoperability are among the challenges and factors affecting administration of data in the modern world [13]. Therefore, the ministry recommends that since the architecture of the system forms the main framework on which digitised land transactions run, a robust and scalable enterprise software architecture with a design guided by interoperability, high availability and e-governance strategy awareness is to be enforced [26].

The automation, interconnectivity, interoperability and creation of a seamless flow and exchange of data amongst all land service providers are crucial [6]. The inter-linkages will enable systems to speak to each other while interoperability facilitates the ministry and the land sector to operate as one strong and stable entity as opposed to different departments operating independently. This is an important step towards full e-governance for Kenya [13].

The benefits of the modern and interoperable system are expected to be enormous, right from users' satisfaction of the services offered to economic development [27]. It is

expected that the modernised land registration system (which brings with it data interoperability) will facilitate a seamless land market and trading system within regions and the world over [13]. Users usually need transparency, efficiency, speed, equitable access, data quality, interoperability and a cost-effective system [5], and therefore, for an effective service delivery, a one-stop shop at the comfort of one's home or office via the Internet is preferred [13].

## 5. Kenya Framework for Interoperability in Land Administration

According to Oukes et al. (2023), the ISO Framework for Enterprise Interoperability (FEI) [20] offers a framework for interoperable land administration systems. FEI considers interoperability in three dimensions, as presented in Figure 4: concerns (red), barriers (yellow) and approaches (blue), where each dimension is subdivided into several essential aspects for removing the barriers and concerns to realise cooperation and interoperability.

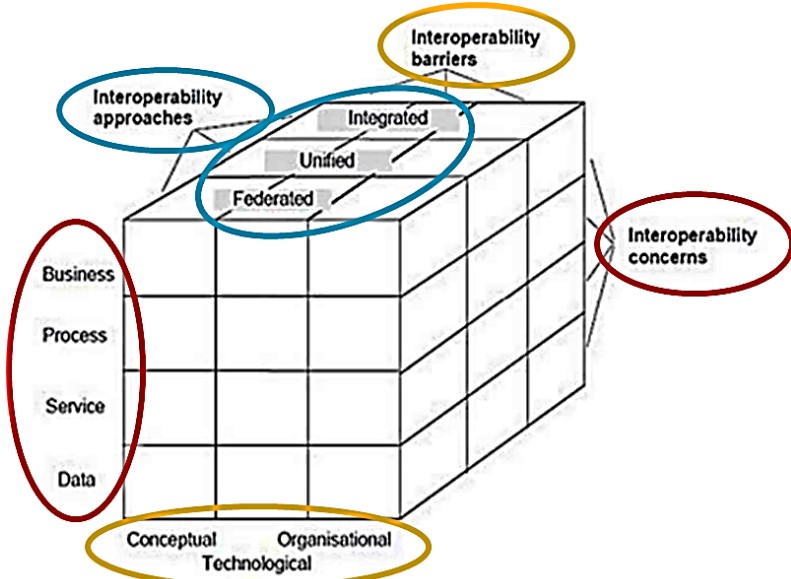

**Figure 4.** ISO 19439:2006 Framework for Enterprise Interoperability [20].

The Kenyan interoperability framework for land administration is proposed in this paper to follow the same principles as in the FEI. This is because similar interoperability issues that are addressed by FEI exist in Kenya too, and the solutions brought by it also favor the Kenyan context.

### 5.1. Interoperability Concerns for Kenya

The LADM profile for Kenya is part of the solution to manage the data, right from collection to storage and distribution. This essentially eliminates all the discrepancies and errors currently being witnessed in the management of LA data [28]. With the introduction of Kenya's e-citizen portal and *Ardhisasa* platform, the demand for land services has risen exponentially [29]. This emanates from the departments or organisations dealing with land to stakeholders, including the general public. Web-based services are a solution that could help to meet the demand [17,28], and, with the improvements in the Internet coverage in Kenya, this is clearly the best option. This implies a critical need to review and update the LA processes by all departments responsible for land data. Automation and digitalisation of the existing manual processes are required. Since different departments have mandates on land information, harmonisation of their goals is necessary to realise business interoperability. This does not necessitate the merging or alteration of organisations [1] but more realignment to serve the common goal. A review of policies, legal frameworks and standards needs to be part of the considerations to address the interoperability concerns [14].

*5.2. Interoperability Barriers in Kenya*

Organisational, technological and conceptual barriers hinder interoperability in implementation and cooperation in LA [14,15]. Restructuring and realignment is necessary for Kenya to remove the organisational barriers in order to entail the policies and regulations that factor in cooperation between departments and the persons responsible for the development and implementation of the linkages. This does not change how the departments are organised nor bring change in their mandates, but more organises the enablement of interaction and cooperation in LA. The use of data models and common technologies is ideal for addressing technological barriers. This could include the adoption of the LADM profile for Kenya [1] and implementation of it [27]. The development of common workflows, processes and tools is critical in eliminating technological barriers. While doing away with conceptual barriers (syntactic or semantic), common standards are to be developed [14]. A starting point could be the adoption of the existing ISO or OGC standards [15]. Utilisation of updated practice manuals such as Kenya's Survey Manual [30] is essential for eliminating conceptual barriers among LA stakeholders.

*5.3. Interoperability Approach for Kenya*

The interoperability approaches presented by FEI in Figure 4 provide a framework for designing an implementation that turns issues and concerns into solutions and barriers into requirements [14]. While ISO provides three approaches (integrated, unified and federated), see [20], an integrated approach is proposed for the Kenyan LA since it is more structured with a standard reference for syntax and semantics and a common language [14] like the LADM profile for Kenya [1]. This will take care of the departments responsible for land information in Kenya within the State Department for Land, the National Land Commission, and the 47 County Governments, among other stakeholders, based on the requirements gathered, which indicates a need for a system that concurrently serves all the land stakeholders [4].

## 6. Mapping the Kenyan LADM Unique Requirements with SDGs

The sustainable development goals (SDGs) were adopted by the United Nations in 2015 as a universal call for action to end poverty, protect the planet and ensure that, by 2030, all people enjoy peace and prosperity [31]. These 17 goals, together with their 169 targets, feature many thematic areas, land management included [6,32]. This being an urgent call, it is expected that all countries [15], developed and developing, action them in a global partnership [33]. As land is an important resource globally, its administration and management are crucial in meeting the sustainable development goals (SDGs), for example, goal 15 and targets 1.4, 2.3, 5.a, 11.1, 15.1 and 15.3 [4,10]. This can be summarised as in Figure 5.

It is therefore critical to investigate the LADM requirements for Kenya and how to map them into the SDGs [32]. Among the many requirements for modelling in Kenya, four issues stood out as the most critical for land modelling: the gender recordation and rights, community land and rights, pastoralist rights and informal occupation and rights [1].

According to Okembo et al. (2023), for rights and gender recordation, attributes on gender type, responsibilities related to land and marriage recognition are identified. For community land and rights, recognition of community ownership and attributes on rights, responsibilities and restrictions are included in the model. In terms of pastoralists, it is important to recognise their way of life through nomadism activities. Attributes on migration patterns, migration corridors, migration periods, grazing areas buffer zones and identification of their stakeholders are incorporated in the profile. Finally, for the informal occupation rights on land, they are to be identified and recorded, with attributes on rights, responsibilities and restrictions. Also, some temporary occupation certificates in the form of a letter of allotment are to be issued to them so as to aid in future relocation and resettlement by the government. The introduction of these attributes requires legislation and process development.

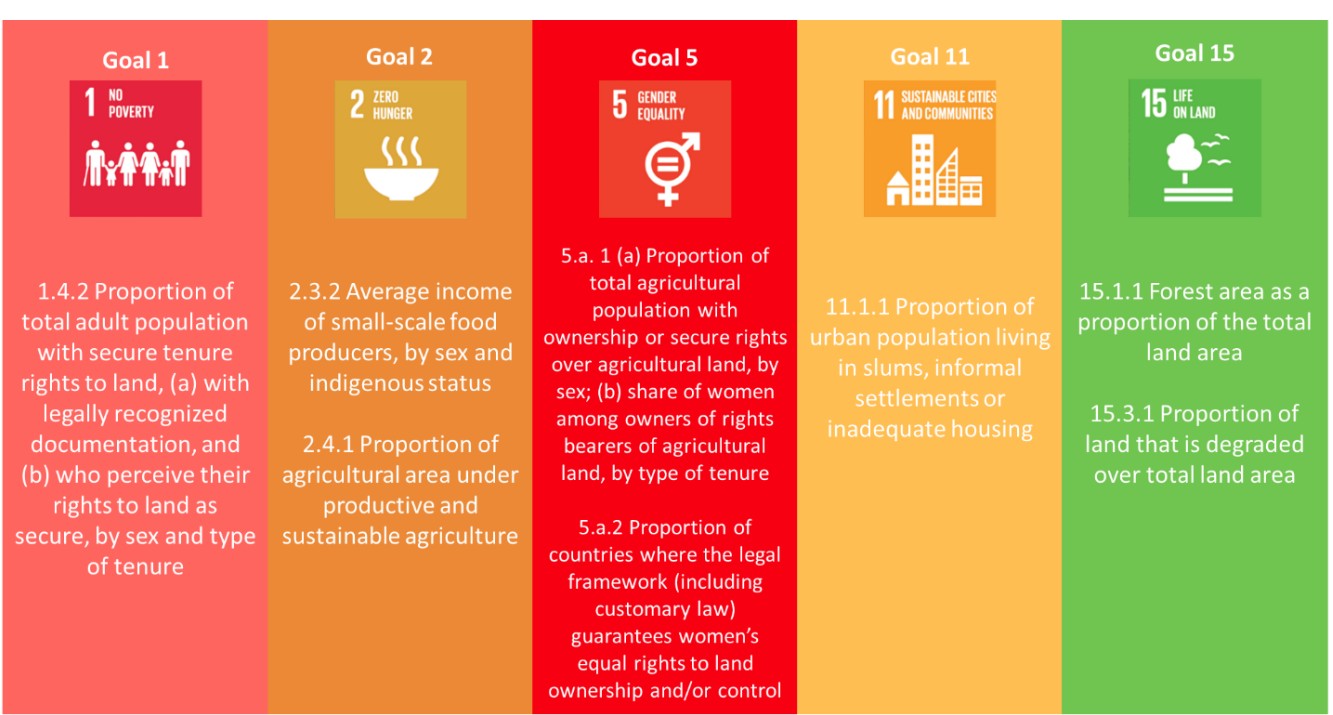

**Figure 5.** SDGs indicators related to advancing LASs, source [34].

Land is at the center of the SDGs and all goals in one way or the other because a country's economy, stability, and sustainability depend on its land [34]. A modern and efficient LAS, which contains geospatially precise representations of land parcels and associated RRRs, is essential for countries to achieve the SDGs [35]. Out of the 17 goals, 5 goals do not speak on land explicitly, being goals 4, 8, 10, 16 and 17 [32]. However, if better argued, they could still be pinned to land somehow. An attempt to match the SDGs with basic classes of LADM and to categorise them into party-centric, RRR-centric and spatial-centric SDGs is carried out by Unger et al. (2019), as visualised in Figure 6.

Gender equality and empowering all women and girls is considered as a goal on its own (SDG 5). The effort cuts across the entire 2030 agenda and reflects the growing evidence that gender equality has multiplier effects on sustainable development, allowing for the eradication of poverty (SDG 1) through its target 1.4: "By 2030, ensure that all men and women, in particular the poor and the vulnerable, have equal rights to economic resources, as well as access to basic services, ownership and control over land and other forms of property, inheritance, natural resources, appropriate new technology and financial services, including microfinance" [37]. This requires gender recordation and is measured by indicator 1.4.2: "Proportion of total adult population with secure tenure rights to land, (a) with legally recognised documentation, and (b) who perceive their rights to land as secure, by sex and type of tenure" [33]. Achieving food security (SDG 2) is carried out through its target 2.3: "By 2030, double the agricultural productivity and incomes of small-scale food producers, in particular women, indigenous peoples, family farmers, pastoralists and fishers, including through secure and equal access to land, other productive resources and inputs, knowledge, financial services, markets and opportunities for value addition and non-farm employment" [37]. This is measured by indicator 2.3.2: "Average income of small-scale food producers, by sex and indigenous status" [33].

Regarding the community land and rights, the model provided a way to recognise the community ownership of land and their management [1,4], hence giving the community members rights over their ancestral land [38]. This is in line with the SDG 2.3, which aims to (by 2030) "double the agricultural productivity and incomes of small-scale food producers, in particular women, indigenous peoples, family farmers, pastoralists and fishers, including through secure and equal access to land, other productive resources and

inputs, knowledge, financial services, markets and opportunities for value addition and non-farm employment" [33]. The focus here is on the indigenous people who are in various communities with a customary way of life. The goal will be measured by indicator 2.3.2: "Average income of small-scale food producers, by sex and indigenous status" [33].

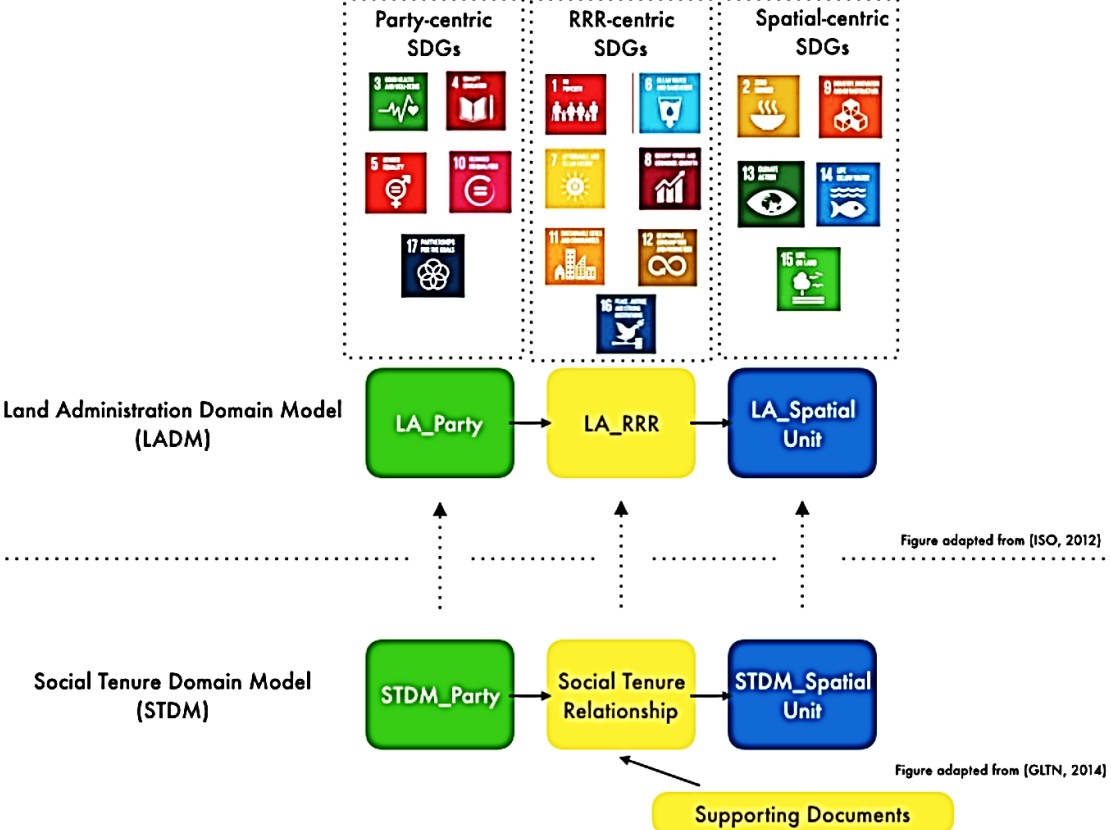

**Figure 6.** LADM as a base to support SDGs, source [36].

The pastoralist rights on land for its grazing, watering and migration are covered also in goal 2.3: "By 2030, double the agricultural productivity and incomes of small-scale food producers, in particular women, indigenous peoples, family farmers, pastoralists and fishers, including through secure and equal access to land, other productive resources and inputs, knowledge, financial services, markets and opportunities for value addition and non-farm employment" [33]. This brings into consideration their way of life and how it supports agricultural production and enhances it for sustaining their livelihoods. Indicator 2.3.1, "Volume of production per labor unit by classes of farming/pastoral/forestry enterprise size" [33], helps us to measure the achievement of this goal.

Finally, for the informal occupation and rights, SDG 11.1 addresses these by stating that "By 2030, ensure access for all, to adequate, safe and affordable housing and basic services and upgrade slums" [33]. In order to achieve this, indicator 11.1.1 requires that the proportion of urban population living in slums, informal settlements or inadequate housing [33] has to be recorded and measured. This means that the parcels' land use and people living on them must be recorded with sufficient information to be able to come up with an estimation in this regard [34].

This relationship between the SDGs and Kenya's four unique issues is summarised in Table 3.

**Table 3.** Matching unique Kenya LADM issues and SDGs.

| Kenya Land Issues | Corresponding SDG | Corresponding Targets |
|---|---|---|
| Gender recordation and rights | 1.4, 2.3 and 5 | 1.4.2, 2.3.1, 5.a.1, 5.a.2 and 5.c.1 |
| Community land and rights | 2.3 | 2.3.2 |
| Pastoralist rights | 2.3 | 2.3.1 |
| Informal occupation and rights | 11.1 | 11.1.1 |

Figure 7 demonstrates the mapping of the SDGs to land based on the LADM packages. Chehrehbargh et al. (2024) grouped all implications of global initiatives on LASs into three main categories: governance, operational environment and sustainability, and mapped the SDGs as presented in Figure 7.

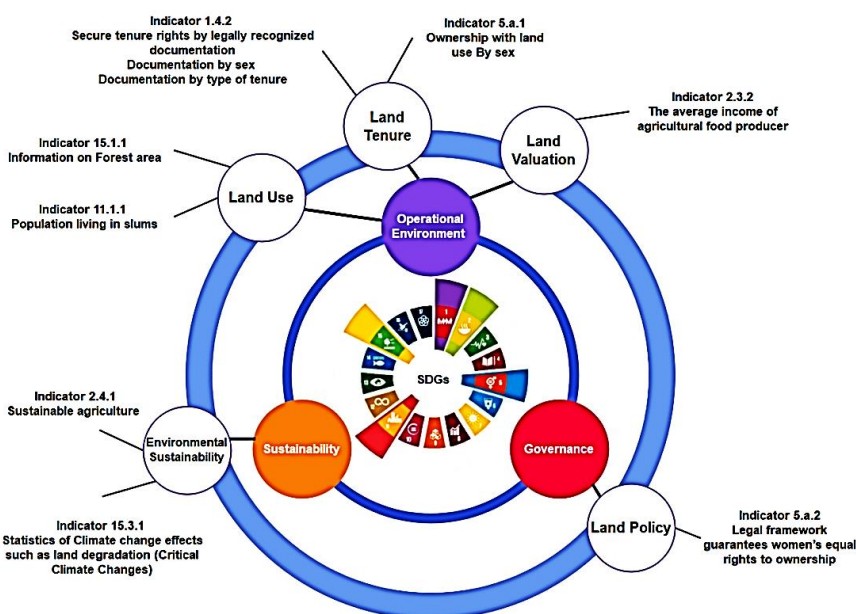

**Figure 7.** Directly mapping land-management-related parameters of SDGs into identified categories. Source [34].

## 7. Colombia and Kenya Testing Review

Kenya undertook a technical test on the STDM based on an FFP approach and Colombia tested their LADM profiles on real-life cases using smartphones, web applications and RTK correction services. In Kenya, Ambani et al. (2017) tested a fit-for-purpose approach to land administration, with a focus on the provision of land titles with inclusiveness for all, where the approach is affordable, fast and 'good enough'. Walking along the boundaries with the stakeholders and using ortho-photos showing most of the boundaries, they were able to delineate land rights. After identifying in the field, the visual boundaries were then drawn in an analogue manner using a pen or 'digitally drawn' using handheld global navigation satellite system (GNSS) devices on top of imagery as shown in Figure 8.

They also, hand in hand with the boundary delineation, captured the people–land relationships, including formal ownership and informal land use, as well as the possession and occupation of lands, including by women. The field test conducted in Makueni County demonstrated that the field data collection and data handling can be carried out quickly, affordably and reliably. This test was carried out by the Institution of Surveyors of Kenya, the National Ministry of Lands, Housing and Urban Development and the Ministry of Lands, Mining and Physical Planning in Makueni County, in close collaboration with software and hardware providers [16].

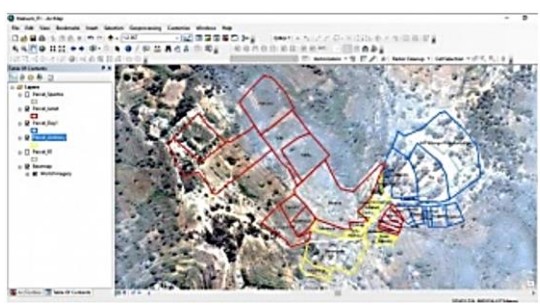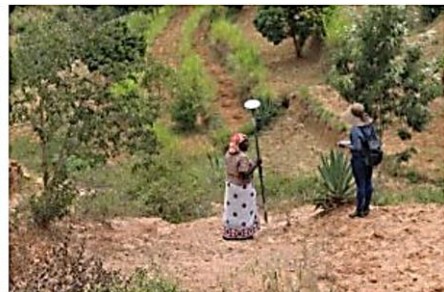

**Figure 8.** Field test in Makueni, Kenya, source [21].

On the other hand, in Colombia, there was a need to modernise land records management from their manual and paper-based processes [22,27]. Several workshops were realised in defining a first version of the Colombian LADM profile, together with the specialists of the National Geographic Institute (IGAC), the Property Registry (SNR), the National Land Agency (ANT) and the Land Restitution Unit (URT). Professionals in charge of the National Spatial Data Infrastructure (ICDE in Colombia) and partially those of the E-Government Strategy of the Ministry of ICT were involved too, both interested in promoting the inclusion of the LADM-COL profile in their normative frameworks [39].

The Colombian profile is based on INTERLIS, which follows a model-driven approach (MDA), with the LADM as the primary standard [40]. This gave birth to the first version of the conceptual country profile for Colombia (LADM-COL), described in UML [40]. The LADM-COL model is modularised and structured around a core or minimum model containing the common elements that define the profile. The model is implemented by the institutions that are responsible for each thematic area of data, specialising it according to their missional needs through specific classes, relationships, attributes, sets of values and constrictions [40].

Then, a pilot for the implementation of the model was planned. To carry this out, a simplified, community-based, standards-compliant methodology was developed with supporting technology in order to register people-to-land relationships in a fast and economically viable way [41]. This was based on the concept of fit-for-purpose land administration and primarily focuses on rural areas [22].

The project's methodology has proven to be applicable and is producing land titles and tenure security for the rural population of Colombia at this moment. The first land titles based on this methodology were handed out in late 2018 [22]. Kenya can learn from Colombia, especially for the rural cadastre, where community land dominates and women's rights need to be recorded.

## 8. LADM Implementation Test in Kenya

The developed Kenyan LADM profile [1] is implemented and a test on its functionality is performed. This includes the conversion of the UML model to a database, migrating existing data to the model, developing a web application, configuration of a field data collection application and undertaking the technical test of the field app. The process is detailed below.

### 8.1. Test Area: Nairobi and Kajiado Counties

Testing of the Kenyan profile was performed within the Nairobi Central Business District (CBD) due to the accessibility and availability of existing data. The data used consisted of 1353 parcels received from the Nairobi County Government for demo purposes only. The editing of parcel geometry and attributes is part of the data. For mapping new parcels and attributes, Kitengela township within Kajiado County, which is south of Nairobi, was used. This was due to the accessibility and limited external interference.

*8.2. Model Conversion to Software*

The first step in the technical implementation is to convert the Kenyan profile, available as a UML model, to a physical environment, that is, a DBMS. In this case, the desktop GIS software geodatabase is used. Since the model has all the classes, attributes, code lists and relationships as well defined, the task here is to move it to a geodatabase format.

The classes are represented as feature classes for the classes with geometry and as tables for the classes with no geometry (with attributes as fields). Special attributes from the unique Kenyan issues, such as marriage type and gender type [1], are created.

Using the Create Relationship Class tool, associations between classes are created together with their multiplicities (cardinality). The code lists are created as domains as presented in Figure 9. The domains are then assigned to their respective attributes using the Assign Domain to Field tool.

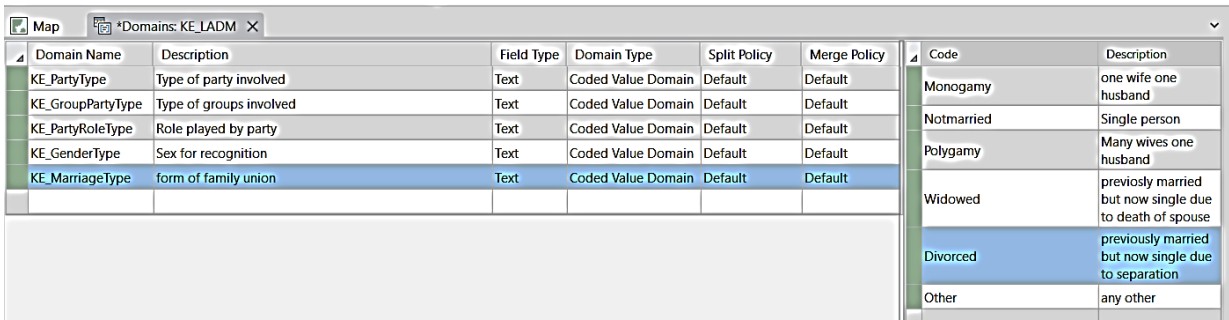

**Figure 9.** Creation of code lists (domains).

A parcel fabric is a capability within the GIS software geodatabase that stores a dataset of connected parcels or a parcel network [42,43]. Each spatial unit type, such as a parcel, is composed of a polygon and line feature class that is defined. Points and connection lines are shared by all spatial unit types and help to maintain the topological integrity and densify the survey network. When a parcel fabric is created, a geodatabase topology is also created to manage the topological relationships between features in the geodatabase, such as polygon overlaps and gaps, and line intersections. The parcel fabric provides a comprehensive framework for managing, editing and sharing parcel data in both a multiuser and single-user environment, and the data will be stored in a geodatabase that can be in the following database management systems (DBMSs): MS SQL Server, PostgreSQL, Oracle, SQLite (mobile geodatabase) or a local file geodatabase [43].

Esri has worked over time with the LADM teams, allowing the parcel fabric to have similarities with the LADM. These similarities can be simply explained because both the conceptual LADM and the physical parcel fabric information models are driven by the same business requirements to describe the same entities, and they both use abstract terminology [43]. Parcel fabric is also conformant with the LADM abstract test suite (level 1–3) [43,44].

Parcel fabric software is used to model the spatial unit package and the surveying and representation sub-package. To perform this, the Create Parcel Fabric geoprocessing tool is used. When the tool completes execution, the spatial unit feature class together with its associated classes, relationship classes and topological rules are created and added to the geodatabase as shown in Figure 10.

All the LADM classes, attributes, code lists, associations and multiplicities in the Kenyan profile are completely created in the desktop GIS software geodatabases as feature classes, fields, domains and relationship classes. To better manage the spatial unit, a feature dataset is created; this contains the associated topology rules and the parcel fabric. The conversion of the special attributes for the Kenyan LADM profile is performed without any problems. A complete data model for Kenya is now in the production state as presented in Figure 11, and the database can now be filled with data.

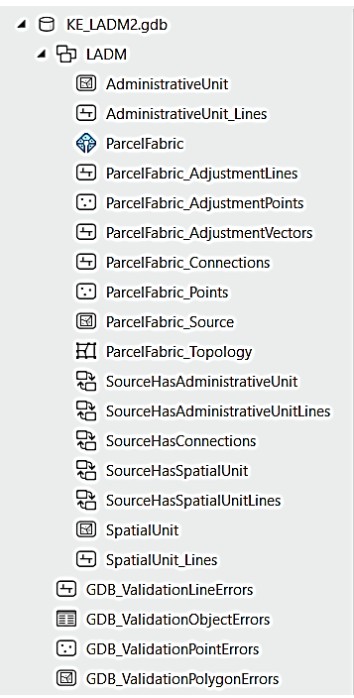

**Figure 10.** Parcel fabric classes and their relationships.

### 8.3. Migration of Existing Data

The migration of existing data is a simple process that does not require prior data cleanup. This allows organisations to move their existing data with any issues into an efficient production system and then evaluate and fix the quality issues later [42]. The data are brought into the database in desktop GIS software through the Append Geoprocessing tool.

After the tool has been fully executed, the spatial units (parcels) are now in the database within the parcel fabric created and can be displayed alongside other data and properties of the geodatabase, such as topologies, as shown in Figure 12.

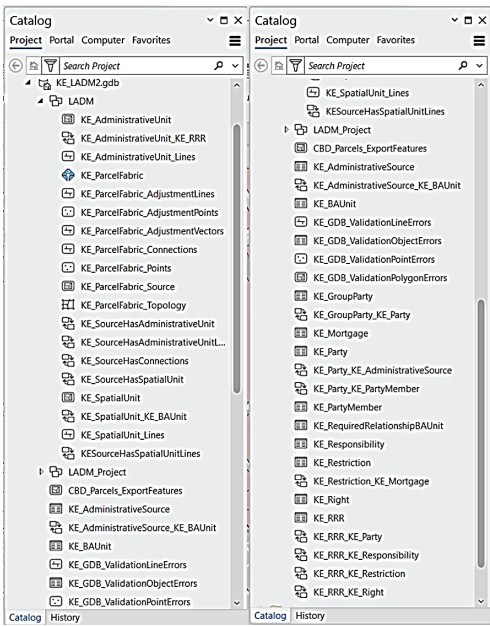

**Figure 11.** Classes (feature classes) and relationship classes (associations) in desktop GIS software.

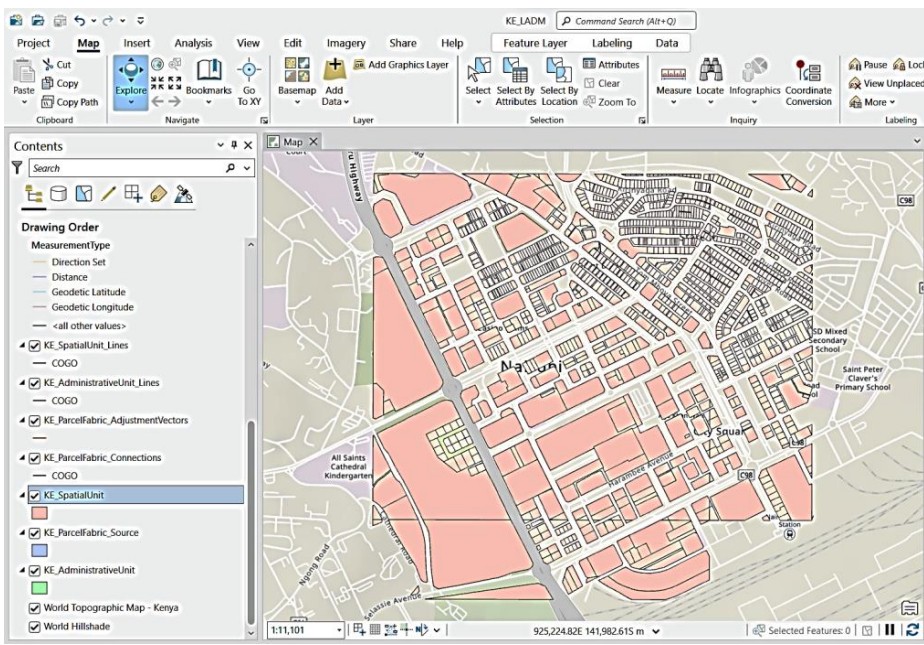

**Figure 12.** Existing data migrated to the parcel fabric.

These data can be queried to show their attribute information and other classes related to it as illustrated in Figure 13.

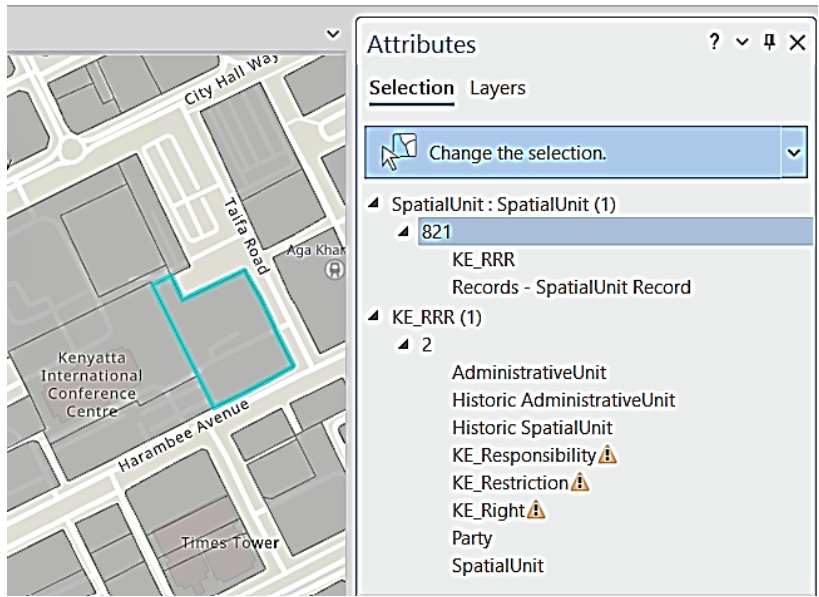

**Figure 13.** Query of spatial unit showing related classes.

*8.4. Web App Development*

Using online GIS software as a web engine, the model and parcel data displayed in desktop GIS software are published as a feature service by sharing as a web layer. The map service can now be accessed through online GIS software content as a feature layer, accompanied by a service definition that stores the metadata for the service as displayed in Figure 14.

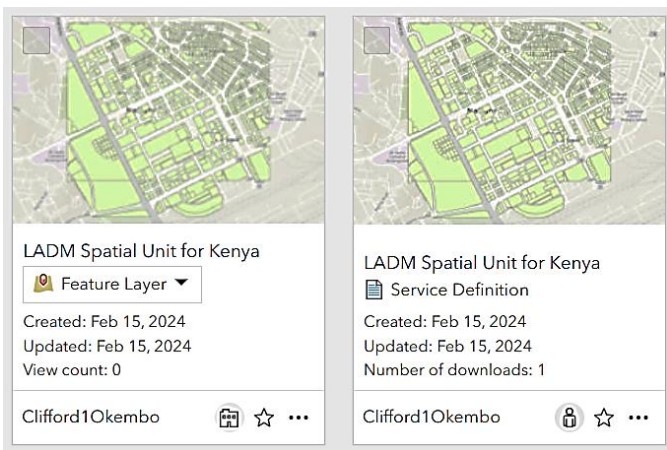

**Figure 14.** Feature layer and its service definition in online GIS software.

Once the service is well set in the online GIS software, setting to enable feature editing (add, update and delete), editor tracking and offline use was carried out as shown in Figure 15. This is to enable all field officers to edit all the features, both geometry and attributes, instantly and, if working offline, to then synchronise once back to where Internet is available.

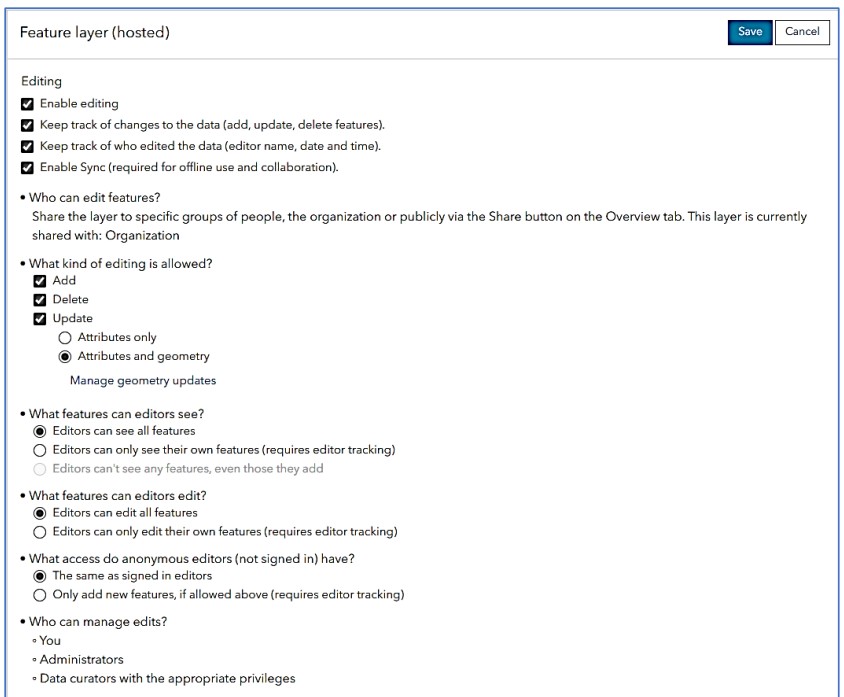

**Figure 15.** Feature layer settings to enable editing over the web.

*8.5. Field App Configuration*

For effective field work, an application (app) running on a smartphone was necessary, just like in Colombia, where the field survey module was based on the Field Maps app (formerly Collector app) and, consequently, the module takes advantage of the cloud infrastructure [22], available with online GIS software. The data published in online GIS software could be accessed through the app with editing capabilities as demonstrated in Figure 16.

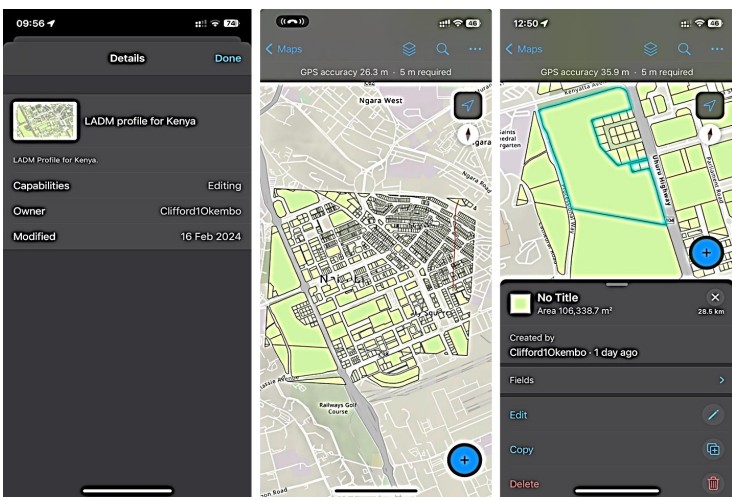

**Figure 16.** Field Maps app (formerly Collector app) configuration.

*8.6. Field App Testing*

Following the previous successful test of the Collector app (now called Field Maps app), which enabled very efficient data collection in Makueni, Kenya in 2017, where the app was used in combination with a GNSS device for sub-meter accuracy via a Bluetooth connection [21]. The same process was followed to undertake the exercise. Confidence in the use of the app was due to its past successes in Kenya and Colombia. For Colombia, the feedback was that the application not only provided support for sophisticated professional surveys but also for more basic surveys that comply with a fit-for-purpose (FFP) philosophy under the supervision of qualified professionals. For these settings, robustness and ease of use were key characteristics. This development undoubtedly opens opportunities to address the many data collection problems that are so common in developing economies [22].

Due to a lack of continuous Internet connectivity, offline files (for both geometry and administrative data) were prepared and the areas for the field test were downloaded in the smartphone.

With them, the field team could map the parcel corners through point averaging, noting the accuracy required, which, for this exercise, was less than 5 m. The attribute information was also entered after the geometry was completed as in Figure 17.

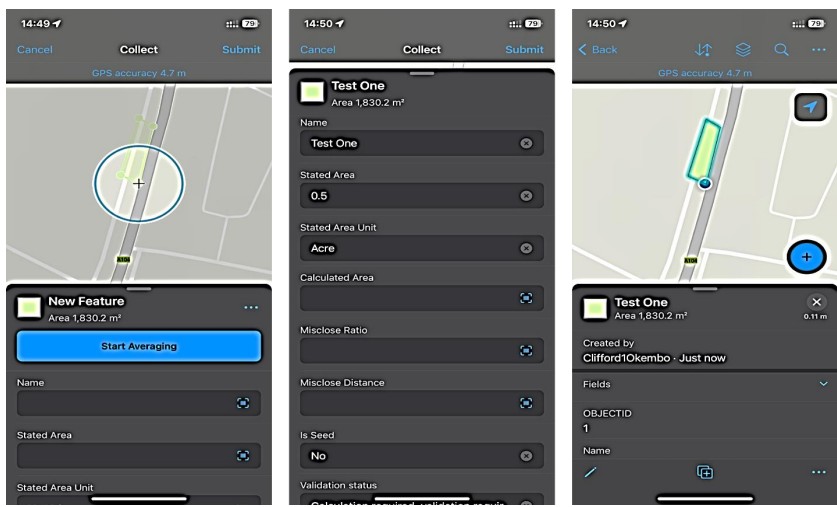

**Figure 17.** Mapping parcel polygons and filling attribute values in the field.

Adjacent parcels were mapped too to demonstrate the possibility of if they were the same and to test the snapping functionalities. Where the geometries did not fit well, correction was to be carried out back in the office using the topology rules already set in the desktop GIS software parcel fabric model.

## 9. Discussions

Data exchange and interoperability within the land administration domain are fundamental for an effective land administration system. By mapping all the data creators and consumers, one is able to tell who generates what data and who needs them, thereby eliminating data creation redundancy and reducing the cost of production while bringing about efficiency. However, many data owners might not be willing to share or provide them as a format or package palatable to the others. This is a great concern that should be addressed before any data interoperability strategy is put in place. Through sensitisation or training in workshops and conferences, together with a comprehensive change management program, this milestone could be attained.

While developing a land administration data interoperability framework for the country and building the foundation with the LADM country profile, there is a need to thoroughly test the framework developed to ensure that it provides the expected outcomes and covers all sectors, areas, functions and workflows intended. This can only be possible if a pilot implementation of the same framework is performed, for example, for one county in Kenya. This could be progressed by expanding to others systematically and in some orders before a national rollout could be carried out. Continuous training, capacity enhancement and attitude need to be strategically planned and executed overtime. Adjustments in the framework can be carried out based on the feedback from the stakeholders involved in each phase or function.

Through the United Nations Agenda 2030, which conceived the SDGs, all the LADM requirements need mapping with the sustainability goals, targets and indicators. While goal five needs to be looked into in totality, more so with regard to its focus on women rights in all resources, other goals have one or a few targets and indicators concerning land-related factors for consideration, monitoring and mapping. Kenya's four unique issues are coincidentally mapping seamlessly with the various SDGs. Regarding gender considerations and pastoralist rights on land, while some SDGs are direct in prescribing what is required, some indirectly point out the areas.

In order to measure, evaluate and report on any of the goals, dedication in the service and collaboration in the implementation are very necessary. The departments in the government are each to map and report on different goals, indicators and targets. Deliberate efforts need to be put in place that support the whole process, but they need to be centrally managed, harmoniously coordinated and reported in a timely manner. Achieving all goals at once might be a disaster, and agility in the process is recommended with the determination to complete it within the time and budget without compromising quality. With land information as the core of almost all SDGs, it is expected that the attainment of Agenda 2030 is feasible. By being more informed and protected due to the security of tenure, the citizenry of the county in question will have not only possessed land but achieved almost all the sustainability goals.

The field work is expected to produce better results with the involvement of the parties owning rights on land, more so for the adjacent parcels. A test on the process with more stakeholders is required. This holds true for the public inspection and workflow implementation. Time constraints and regulatory requirements made these not possible at this stage; however, they are planned to help to obtain the acceptance of a wider audience, the public and the other stakeholders. It is expected that an actual test with actual intended users will help to validate the technical successes already achieved with this test. On the other hand, the actual integration with an external database, especially through the Integrated Population Registration System (IPRS), is important in attaining an integrated land management system. While the LADM proves that integrating a cadastre and registry

is no longer a myth, the addition of other databases will help Kenya to move in the direction taken by the Dutch in their key registers.

There is a need to undertake a public inspection where stakeholders straight from the community are involved to verify the data collected. This includes boundaries and the administration data collected, including the photographs and signatures of parties. The professionals in the land sector, such as surveyors, registrars and land administrators, are to be involved. During this stage, processed results are presented to the community in a public forum for approval. This is where signatures are collected as validation of the results, indicating agreement between parties [22]. This could not be achieved due to the limited time and the bureaucratic process to obtain the permission to have the exercise performed. It is, however, planned for in the near future.

Integration of the land data with external databases is necessary. This is because the LADM profile for Kenya is meant to integrate with external data sources such as data from the Registrar of Persons database [1]. This is to facilitate data integrity and to avoid duplication of data entry. Data from external databases such as Kenya's IPRS [45] are integrated with the LADM system in order to test the interoperability of different data sources into the model. Kenya's department of immigration under its National Registration Bureau has, among its Vision 2030 objectives, a population registration and immigration services program. The program's objective is to establish and operationalise an IPRS for Kenya [25]. The IPRS is committed to developing an application programming interface (API) that is to support the integration of the IPRS to all the government systems and databases. The IPRS database combines primary registration, such as birth, national ID, passport and alien ID, with secondary registration like tax registration, telecom SIM registration, driving license, national health insurance number, national social security number and GIS positioning of place of business and residence [46]. While IPRS has been integrated with the mentioned secondary registrations, it gives an indication of the possibility with land registration. It is therefore paramount that the same integration be carried out for the land registry to serve as a reference and also validation of KE_Party details. The entry or collection of party information will not be necessary since it can be drawn from the IPRS and validated by attaching the photos of the national ID card or passport.

Piloting the system and the workflows therein in a staging environment enables testing processes in land within the model to ascertain whether the model supports transactions such as parcel search, transfer of land rights or parcel subdivision. To carry this out, queries using some structured query language (SQL) or over-the-web services [23] using progressive languages like JSON are necessary. This shall support workflows such as spatial unit (parcel) search to obtain the status of the land, parcel subdivision of amalgamation and the whole parcel registration process producing a title deed as an output.

For ease of implementation, a testing environment that supports the validation of the whole exercise is crucial, starting with implementing the LADM model in a software, configuring the database through parcel fabric, making it available online through an API, collecting and editing data while in the field and transferring them to the cloud. It is recommended that a test for the whole model be completed, managed under the parcel fabric, which would help to model many functions that could be undertaken on a desktop software. Considerations for a web- and mobile-based application giving a one-stop-shop for land matters are essential.

## 10. Conclusions and Recommendations

Exchanging land data from one department to the other is very crucial. Interoperability of the same is equally essential. For this to be effective, mapping of the data creators or generators is one of the key tasks. Identifying the consumers for the same works well in determining who needs the data produced by who. It is at this point that the need for an interoperability framework becomes necessary. Using the ISO Framework for Enterprise Interoperability standards and the LADM country profile forms a solid foundation for an

interoperability framework for a country, more so in land administration. Addressing all the interoperability concerns, removing all the interoperability barriers and determining the best interoperability approach becomes a solution.

Land resources and their information are at the center stage of society sustainability. Forming a core of the SDGs, mapping land resources, their ownership, utilisation, value and protection is very important. This enables the tracking and monitoring of the attainment of the sustainability goals and more so mapping the realities on the ground and the SDG targets and indicators. Promoting the different groups in the society—women, pastoralist, indigenous and informal owners—to have some part to play in areas that help to reduce, if not eliminate, poverty, hunger and diseases, among other global concerns, requires proper land information management: collection, recordation, storage, editing and sharing.

Systems, such as ArcGIS, for desktop, web and field use have proven to offer better solutions to implementing the LADM country profile for Kenya, and this could be true in other countries as well, more so the developing ones. While they support existing data, new data could also be collected, in either online or offline mode, depending on the Internet connectivity in the area in question. However, the choice of system for adoption should be carried out very carefully and consider other factors, especially with regard to the ICT guidelines and standards for that country.

The objectives of this research were met and the research questions were answered. Using the Kenyan profile, its conversion to software, utilisation over the web application and testing of specific attributes in the field make this paper original and a foundation for further findings in this thematic area.

**Author Contributions:** This research is a result of the collaboration and contribution of all authors. Conceptualisation, C.O. and C.L.; methodology, C.O. and C.L.; validation, J.Z. and J.M.; writing—original draft preparation, C.O.; formal analysis, J.M. and J.Z.; writing—review and editing, C.L., J.Z., J.M., D.K. and C.O. All authors have read and agreed to the published version of the manuscript.

**Funding:** This research received no external funding.

**Data Availability Statement:** No new data were created or analyzed in this study. Data sharing is not applicable to this article.

**Acknowledgments:** We acknowledge Amir Bar-Maor for his support with the use of Esri's parcel fabric and the linkage with LADM.

**Conflicts of Interest:** The authors declare no conflicts of interest.

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
