# Peer review of "A Land Administration Data Exchange and Interoperability Framework for Kenya and Its Significance to the Sustainable Development Goals"

_land, doi:10.3390/land13040435_

Round 1

Reviewer 1 Report

Comments and Suggestions for Authors

A Land Administration Data Exchange and Interoperability 3 Framework for Kenya and its significance to the SDGs

The authors done a good job and achieved the proposed objectives in a sound way

In the abstract Add some results what your study finds something new.

Arrange the keywords in alphabetic order.

Line 103, in methods, authors insert same reference for two tiles. Authors needs to be rechecked.

Improve the resolutions of the images.

In methods, why authors insert the screenshot of software used? Delete all images and insert he paragraphs instead.

Discussion in too long, authors need only discuss their study objectives, how they achieved, and results. May compare their study with other studies.

Some of the references are incomplete. Authors needs to check all and provide all the information,

The format of the paper/references is totally out of order.  

Comments on the Quality of English Language

Moderate editing of English language required

Author Response

Responses uploaded as a word document.

Reviewer 2 Report

Comments and Suggestions for Authors

The submitted manuscript tries to cover many different research areas in land administration, such as data integrity, interoperability, dissemination, LADM implementation and relations of LADM with Sustainable Development Goals (SDGs). However, reading through the paper it does not sufficiently contribute to any of the said topics. I would like to recommend that authors start with the developed LADM country profile in Kenya and select one of the following research areas such as: LADM as a support for integrated land administration in Kenya, implementation of LADM Kenya country profile for land registration (this is referred to the conducted test by using ArcGIS in the paper) or to examine how does the developed Kenya country profile addresses the SDGs (which can be linked with LADM). Choosing to focus on one of the said topics makes way to discuss findings of one research with research challenges of other topics.

Furthermore, the manuscript does not provide sufficient literary sources regarding wide range of topics which are addressed. I suggest examining recent proceedings of LADM workshops, some of the recent papers thematically related to development of LADM and land administration in general. As few starting points I recommend examining the following sources: (and further sources in these could be helpful):

1. UN-GGIM Framework for Effective Land Administration; United Nations: New York, NY, USA, 2020.

2. UN-IGIF United Nations Integrated Geospatial Information Framework Part 1: Overarching Strategy 2023.

3. FAO; UNECE; FIG Digital Transformation and Land Administration - Sustainable Practices from the UNECE Region and Beyond 2022.

4. Križanović, J.; Roić, M. Modeling Land Administration Data Dissemination Processes: A Case Study in Croatia. IJGI 2023, 12, 20, doi:10.3390/ijgi12010020.

5. Vranić, S.; Matijević, H.; Roić, M.; Vučić, N. Extending LADM to Support Workflows and Process Models. Land Use Policy 2021, 104, 105358, doi:10.1016/j.landusepol.2021.105358.

6. Kara, A.; Lemmen, C.; Van Oosterom, P.; Kalogianni, E.; Alattas, A.; Indrajit, A. Design of the New Structure and Capabilities of LADM Edition II Including 3D Aspects. Land Use Policy 2024, 137, 107003, doi:10.1016/j.landusepol.2023.107003.

7. Alvarez, G.; Bernal, L.; Albarracin, J. Interaction of Land Objects from LADM Models to Improve Citizen Services. In Proceedings of the Protecting Our World, Conquering New Frontiers; FIG: Orlando, Florida, USA, June 28, 2023.

8. Proceedings of LADM workshops

Reading through the paper I have made quite a number of notes which should be considered in revised/resubmitted version of the manuscript:

- Abstract should be more concise and should simply reflect identified research gaps and achieved results.

- I recommend omitting company and name of the employed software as some readers might perceive the paper as marketing material of the certain company and its products.

-  Keywords such as “Kenyan LADM profile” should be split into keywords such as “Kenya”, “LADM”, “Country profile”.

- There is inconsistent use of terminology, terms such as information provision, data exchange, sharing could be grouped under term data dissemination as this term is used when describing land administration by UNECE, Framework for Effective Land Administration etc. If any other term is selected then it would be helpful to use it throughout entire manuscript.

- Lines 38-39, explaining functional obstacles in Kenyan land administration system (LAS) should be included in the Introduction instead of just acknowledging it. Readers who do not live in Kenya might not be aware of the present issues.

-  Line 45, Siloed data management is global issue, and it should be cited that it is recognized by institutions such as FIG (e.g. FIG publication no. 80).

- Line 62, what are LA units? I think better fitting terms would be LA authorities/users/actors.

- Line 68. KRA abbreviation is not explained in this part of the paper. Take care in using full names before abbreviations. Another examples include line 225 (LIMS) and line 686 (IPRS).

-  Lines 67-68 should be complemented with more sources.

-  Figure 1 is not related to the objective of paper; moreover, Dutch system is mentioned only in Discussion as an example of what would Kenyan system strive to become.

- Line 86: I suggest formulating questions into:

1) What are the use cases or user requirements in Kenyan LAS?

2) Who are the involved stakeholders?

3) How to achieve interoperability? (Current state versus improved state)

4) The fourth question is redundant as it can be discussed as an objective of interoperable LAS.

5) If Kenyan LADM country profile is a starting point, what is meant by its functionalities?

- There are twelve sections in this paper, not eleven (References are number 12 in the submitted version of the paper).

- Line 114: I would not suggest using the term such as “data originator”, I suggest using term such as authority. Furthermore, when mentioning LA functions for the first time, it would be adequate to name them (Tenure, Value, Use, Development).

- Line 135, Should not this be in background or introduction?

- Line 145, Why mentioning QGIS here, when the presented work is depicted in ArcGIS?

- Lines 162 and 163 contain unfinished sentence.

- Lines 141-170: Steps 5 to 9 should be reworked as they can be replaced by steps such as development, testing, deployment etc. Configuration of application tool can be considered as its development since you develop application based on Kenyan profile and then configure it based on Kenyan profile as well.

- Line 174, I suggest avoiding term like “generate”, a more fitting term would be “to register”.

- Line 178-185; Tables should have boundaries. I recommend using journal template for table design.

- Line 188; How is this concluded?

- Lines 218 to 220 should be complemented with more sources.

- Lines 218 to 230 would fit more in the Introduction.

- Lines 230 to 242 must be complemented with more sources, especially those regarding practices in other jurisdictions.

- Line 252, ISO 11534 stands for Iron ore, not for FEI

- Line 262 – Has any other research recognized these errors you mention?

- Line 267, Nowadays, when almost all of our activities are conducted in the electronic environment it is natural to expect that services are best option for Kenyan LAS.

- Section 5.2. now introduces processes in this paper and the need for standardization of workflows and tools as well. This seems like totally another topic and is disconnected from the rest of the paper. Furthermore, regarding identified parameters of Kenyan interoperability framework, it should be clear which of them are addressed in paper and which are addressed either in other sources or should be addressed in future research.

- Line 294 – What are the methods you mention? Explaining available methods would provide more clarity into your choice of method for achieving interoperability.

- Section 6 is disconnected from the rest of the paper. I suggest discussing LADM role in achieving SDGs either in discussion or introduction.

- Section 7 is not required. Explaining similarities and differences with Colombian case should be mentioned in Methodology as it might be used to explain scientific contribution of this paper’s results and analysis. Moreover, the focus of the paper is Kenya.

- Line 467. In this line you mention “parcel search workflow”, however in the following text there is not a single mention of parcel searching, instead there is description of parcel measurement using the application on a mobile phone.

- Section 8 resembles a manual for using the employed software solutions. I suggest developing a workflow diagram which would depict use case, requirements, data, results, challenges etc.

- Line 531, Figures like Fig. 14, really should be just described in text.

- Lines 697-701; Means of presentation (projector, printed papers etc.) should not be part of a scientific article at least in this topic.

- Line 727, How is it concluded that JSON could be useful for integrating. Showcasing some of the results from other sources on this topic might prove your point.

- Line 643 and line 740 are firstly similar, and secondly are written as conclusions of extensive research. Even though data integrity, interoperability and dissemination are crucial, I would suggest avoiding formulation of sentences like this in the paper, especially without references or clear results to complement them.

Comments on the Quality of English Language

I would strongly suggest using more linking words to connect sentences in order to improve reading quality for potential readers of the paper. Additionally, the manuscript could benefit of expert proofreading as some of the sentences are quite long and might require correction of grammatical tenses.

Author Response

Response uploaded as a word document.

Reviewer 3 Report

Comments and Suggestions for Authors

The paper addresses an important issue: How to foster data exchange between public bodies and how to enable interoperable systems, especially between land related tasks. The connection to the SDGs is a good argument for the implementation, it might not be as relevant for selecting a technical and organizational approach.

While the topic itself is relevant and fits the scope of the journal well, the text itself needs some improvements. The authors quickly introduce Ardhisasa and quickly shift to the lack of framework for sharing data. A question that is not answered until the end of the paper is the availability of digital data in general. However, necessary digitization could be a motivation to use such a framework because then contradictions between organizations could be solved during the digitizing process (at least severe ones).

Except for the lack of a framework, the authors also do not provide details about current limitations or problems. Some potential are mentions (e.g., the mining cadastre) but without knowing the current problems it is difficult for the reader to see the benefits.

Another problem is that the proposed solution is only an overlay of different data. This is the first step of data exchange (make it visible) but it is far from interoperability.

In general, the lack of background information on the current status continuous throughout the paper and prevents the reader to easily see what the gain of such a system for Kenya is.

Critical points:
Fig. 1 can be part of the paper as an example for a framework but just to show that spatial and non-spatial data can be connected in such a system it not really helpful because it lacks arguments why specific connections need to be done (e.g., between vehicles and the cadastre). It could be sufficient to tell a short story where such a connection is relevant.

The content of the paper and the title do not match. The title does not say anything about the field test. In Fig. 2 the testing is a third of the procedure and in the text it covers 12 of 26 pages. The connection to the SDGs, on the other hand, is in the title but only one sub-task in Fig. 2 and 3.5 pages in the text.

Lines 235 ff the authors argue that the benefits of a modern and interoperable system include a worldwide land market. How does this fit the findings of The World Bank on land grabbing? Usually national land administration systems make land grabbing easier, especially it it is open to a worldwide market ...
In the list of user needs maybe safety and transparency are missing. It could also be interesting, which users prioritizes which needs

In the section on interoperability barriers, I am missing two potential obstacles: Lack of availability of digital data and lack if political/administrative will.

Argumentation needs to be improved in some parts, e.g.
- lines 131-132: It is unclear how the LADM requirements can lead to the identification of the SDGs.
- lines 180-183: It is unclear how the inclusion of external data sources can help eliminate duplicates and improve efficiency. Can you provide an example or two?
- lines 261-262: How can the LADM profile eliminate discrepancies and errors when data collection might be done offline (see test section) ...
- lines 270-271: What if the harmonization of goals is not possible for two departments (e.g., nature conservation and economic growth)?
- lines 554-555: Why does the migration not require data cleaning? If the data is wrong (e.g., there has never been a specific right for a specific person on a piece of land), how will it be corrected (eliminated) later?

Minor details:
- lines 36-38: Why is it not intended for land owners as a major group of LA stakeholders? Later they are included ...
- lines 41-43: This also produces inconsistencies
- lines 79-82: This paragraph is unclear. It lacks information on the overall research and how the paper fits into this. I am also not sure whether the paragraph is necessary at all.
- line 103: The references probably should have been "(Okembo, et al., 2022; 2023)" or "(Okembo, et al., 2022; Okembo, et al., 2023)"
- Figure 2: The process is linear, why this complex arrangement in the figure?
- line 123, "Kenya's unique requirements": The word "unique" is not necessary because this is already included in "Kenya's requirements".
Tables 1 and 2 could be figures or lists, they do not really look like tables
- line 187: The heading is ambiguous. It could be the requirements for interoperability (by public administration) or the requirements of interoperability (on public administration). In other words, which is the direction of thew discussion?
- line 197: What are Kenya's Vision 2030 goals? Afterwards only laws are listed but no goals.
- line 206: Is the MLPWHUD the same as the Ministry of Land and Physical Planning? If yes, why are there different names, if not, what is their relation and potential overlap?
- lines 488-489: What is meant by marriage type and gender type? What are the peculiarities in Kenya?
- Figure 14: There is a lot of empty space and the text is difficult to read
- Figures 10 and 12: Text is difficult to read.
- lines 564: Please describe what other data are displayed in Fig. 18.

Author Response

Responses uploaded as a word document.

Round 2

Reviewer 1 Report

Comments and Suggestions for Authors

Authors follow the comments. I am happy with these changes.

Comments on the Quality of English Language

Minor editing 

Author Response

Thank you, happy to have worked with you.

Reviewer 2 Report

Comments and Suggestions for Authors

The revised version of the submitted manuscript is indeed improved when compared to the first version.

There are still few minor issues which should be revised in order to fully improve the overall quality of paper:

1) In the first version of the manuscript in line 89 the authors set the question 5 as: "How does the developed LADM profile for Kenya function? In the review comments I have asked the question as to what is meant by Kenyan LADM country profile functionalities. It was not my intention to suggest formulating the question into "If the Kenyan LADM country profile is a starting point, what is meant by its functionalities?". Perhaps the question should be formulated as to does Kenyan LADM country profile function in the real world implementation.

2) In your response you have mentioned that you adopted term "authority" instead of "originator", however, the captions of Table 1 and Table 2 still use term "originator". I suggest checking the paper in case of other terminology inconsistencies.

3) Figure 4 still has incorrect name of the ISO standard in its caption. ISO 11534 stands for "ISO 11534:2006 Iron ore - Determination of tin". I suggest checking the source of the Figure 4 and correcting it in the manuscript.

4) Regarding the Section 6: Mapping the Kenyan LADM Unique Requirements with SDGs, I suggest checking and adding reference from LADM 2023 Workshop, namely "SDG Land Administration Indicators based on ISO 19152 LADM" by authors M. Chen, P. van Oosterom, E. Kalogianni and P. Dijkstra as it might be related to the content of this section and establishing links of Kenyan LADM profile with SDGs. The conference paper can be found conference website: https://www.gdmc.nl/3DCadastres/workshop2023/programme/3DLA2023_paper_F.pdf

Author Response

The response is uploaded.

Reviewer 3 Report

Comments and Suggestions for Authors

The responses to the comments was sufficient.

Author Response

Thank you for your great work.